# Effect of Internal Donors on Raman and IR Spectroscopic Fingerprints of MgCl_2_/TiCl_4_ Nanoclusters Determined by Machine Learning and DFT

**DOI:** 10.3390/ma15030909

**Published:** 2022-01-25

**Authors:** Maddalena D’Amore, Toshiaki Taniike, Minoru Terano, Anna Maria Ferrari

**Affiliations:** 1Dipartimento di Chimica, Università di Torino, Via P. Giuria 5, 10125 Torino, Italy; 2Graduate School of Advanced Science and Technology, Japan Advanced Institute of Science and Technology, 1-1 Asahidai, Nomi 923-1292, Ishikawa, Japan; taniike@jaist.ac.jp (T.T.); terano@jaist.ac.jp (M.T.)

**Keywords:** polymerization catalysis, machine learning, nanoclusters, DFT, Raman spectrum, IR spectrum, Lewis bases

## Abstract

To go deep into the origin of MgCl${}_{2}$ supported Ziegler-Natta catalysis we need to fully understand the structure and properties of precatalytic nanoclusters MgCl${}_{2}$/TiCl${}_{4}$ in presence of Lewis bases as internal donors (ID). In this work MgCl${}_{2}$/TiCl${}_{4}$ nanoplatelets derived by machine learning and DFT calculations have been used to model the interaction with ethyl-benzoate EB as ID, with available exposed sites of binary Ti${}_{x}$Cl${}_{y}$/MgCl${}_{2}$ systems. The influence of vicinal Ti${}_{2}$Cl${}_{8}$ and coadsorbed TiCl${}_{4}$ on energetic, structural and spectroscopic behaviour of EB has been considered. The adsorption of homogeneous-like TiCl${}_{4}$EB and TiCl${}_{4}$(EB)${}_{2}$ at the various surface sites have been also simulated. B3LYP-D2 and M06 functionals combined with TZVP quality basis set have been adopted for calculations. The adducts have been characterized by computing IR and Raman spectra that have been found to provide specific fingerprints useful to identify surface species; IR spectra have been successfully compared to available experimental data.

## 1. Introduction

In real heterogeneous catalysts the complexity of the sytem is crucial in determing activity, selectivity and life-time, that represent the main parameters defining the performance of the catalyst; hence togheter with active species, dopants, promoters or modifiers need to be carefully chosen and their role fully understood. With regard to MgCl${}_{2}$ supported Ziegler Natta catalysts (ZNC) for olefin polymerization, the electron donors (Lewis base molecules) have been providing that supporting activity since 1960s with significant increase of both catalyst productivity and stereospecificity [1,2,3,4].

This very diffuse practice has been continously improved during last decades by industry mostly in a trial and error process.

The coadsorbtion of electron donors (generally bidentate organic molecules like diesters, diethers, and dialkoxysilanes) determine the local environment of the transition metal, behaving as ancillary ligand(s) in molecular catalysts able to enhance their regio- and enantioselectivity at the propene insertion in a defective context. They also determine the molecular weight distribution of the polymer, the distribution of active sites, the morphology of the catalyst [1,5,6,7,8,9,10,11,12,13,14,15,16,17,18,19,20,21] and the properties of the produced polymers. The industrial research in the last decade was devoted to the optimization of new couples internal donor/external donor (ID, ED) [20,21] because generally pairs of electron donors are used [6,9,22,23].

Both experimental [24] and computational [18,25,26,27,28,29,30] research has been performed to get insights in the structure of ZNC at the atomic level, with main attention to the place and ways of binding of the ID, ED and their effects on the structure of the active sites.

Several contrasting hypotheses have been formulated about the role of the IDs. Some researchers claimed that the ID coordinates to a vacancy of the titanium centers, thus forming new active and stereo-specific sites [31,32,33,34]; it is also supported by the formation of homogeneous complexes between TiCl${}_{4}$ and many Lewis base molecules. The other opposite hypothesis recall the historical one: the ID only interacts with the MgCl${}_{2}$ support, indirectly determining the place and the distribution of the TiCl${}_{4}$ moieties on the MgCl${}_{2}$ surfaces and preventing the formation of aspecific sites [31,32,33,34,35,36,37,38]. Nowadays the most trusted and accepted hypothesis for ID, is the one known as the “coadsorption model”: ID would coordinate to the MgCl${}_{2}$ surfaces close to the titanium active sites, thus influencing their steric and electronic properties with no direct bonding [21,25,26]. We can assume a total consensus of scientific community on the nanostructured and disordered nature of the active δ form of MgCl${}_{2}$ exposing surfaces and defective sites to adsorption of both TiCl${}_{4}$ and internal donors to form Ziegler-Natta precatalyst. The hypothesis supported by experimental investigation, mostly by TEM microscopy, encountered a lot of limits in characterization due to these nanosize and disordered features, heterogeneity of sites, dilution of the active centres, air sensitivity. Theoretical and computational chemistry provided crucial insights in δ-MgCl${}_{2}$, made feasible the modelling of each component and the analysis of different catalytic sites [39,40,41].

Together with some experimental inconsistencies, DFT studies have definitely set aside models obtained by Molecular Mechanics investigations on binary systems TiCl${}_{4}$/MgCl${}_{2}$ and Ti${}_{2}$Cl${}_{8}$/MgCl${}_{2}$ [42], after puzzling DFT predictions of adsorption energies of Ti${}_{x}$Cl${}_{y}$ species on regular MgCl${}_{2}$ surfaces and thanks to more recent reliable descriptions of the morphology of the δ-MgCl${}_{2}$ particles [43,44,45,46,47,48,49,50,51]. Theoretical calculations on MgCl${}_{2}$ bulk and surface structure, exploiting periodic Density Functional Theory approaches including London interactions, pointed out that MgCl${}_{2}$ crystals in the absence of adsorbates mainly expose the (104) or other pentacoordinated surfaces, whereas the picture drastically changes in the presence of adsorbates (i.e., electron donors); in fact, regular MgCl${}_{2}$ crystals expose the (110) surfaces at a relatively large extent when they grow in the presence of Lewis bases (small molecules such as methanol, ethanol, and dimethyl ether) or in presence of alkoxy silanes employed as external donors [30,52]. Thanks to Bravais’ law it was possible to identify the most stable surfaces in terms of (surface) Gibbs Free Energy: (107), (012), (101), (015) and (110) ones [53]. The combination of a careful structural analysis of the supporting δ-MgCl${}_{2}$ nanocrystals by synchrotron X-ray total scattering [40] with the definition of Wulff’s polyhedron of MgCl${}_{2}$ crystallites suggested that ball-milling of naked MgCl${}_{2}$ produces a larger total surface area and modifies the relative extension of the surfaces, too. That would increase the stability of the lateral surfaces exposing strongly acidic tetracoordinated Mg${}^{2}$${}^{+}$ sites (i.e., the (110), (012) and (015) ones) at the expenses of the basal (001) one. The polar plot identifyed the presence of possible defective edges involving the (110) surface [41]. Inelastic Neutron Scattering (INS) spectroscopy [54], able to sample all the first Brillouin zone (FBZ) and thus predicting the dynamical behaviour of materials, proved to be a powerful method to understand nano-sized systems and measure the degree of disorder of δ-MgCl${}_{2}$ in terms of both translational and rotational contribution. A step forward to determine the morphology of platelets, with dimensions comparable to those of primary particles typically adopted in industrial catalysis (2–3 nm), have been made by us thanks to the adoption of a genethic algorithm; the calculations have been performed both in presence and in absence of TiCl${}_{4}$ [55]. Differently than in the case of naked MgCl${}_{2}$ where the evolutionary plots allowed us to identify structures where the majority of sites are pentacoordinated Mg${}^{2}$${}^{+}$ sites, in presence of TiCl${}_{4}$ (in a ratio 50/3 MgCl${}_{2}$/TiCl${}_{4}$) couples of tetracoordinated sites are present after the reconstrution of nano-particles. That occurrence represents the topological requirement for the formation of octahedral Ti species that will start the catalytic process. That “motiv” confirms the presence of inter-surfaces edges as possible favorable place for stereo-selective active sites as also reported in previous computational studies adopting traditional Density Functional methods [29,41]. FT-IR spectroscopy of probing carbon monoxide also was used to estimate the presence and the relative extention of MgCl${}_{2}$ surfaces [53,56]. Very recently, vibrational Far-IR and Raman simulations have been carried out on the nanocluster models identified through a non-empirical structure determination to discuss the effect of the binding of various precatalytic complexes Ti${}_{x}$Cl${}_{y}$ on IR and/or Raman features. IR features do not allow to clearly identify the presence of a certain adduct but in Raman a drastical change occurs after TiCl${}_{4}$ adsorption. The spectroscopic features identified are specific of TiCl${}_{4}$ adduct on (110)-like tetracoordinated Mg${}^{2+}$ sites and different in shape and frequencies from those obtained in the case of the dimeric adduct on pentacoordinated sites. Hence Raman spectroscopy provides clear fingerprints to recognize each adduct on defective positions of nanoplatelet of Ziegler-Natta catalysts.

In this work, we make a step forward towards complexity and pass to ternary donor MgCl${}_{2}$/TiCl${}_{4}$ systems; we looked to different way of binding of ethyl benzoate (EB) in relation to available Mg${}^{2}$${}^{+}$ sites of attachment and respect to positions of Ti${}_{x}$Cl${}_{y}$ complexes, thus following the still open hypotheses on the mechanism that donors adopt for affecting polymerization catalysis. Differently than previous papers, mostly addressing perfect surfaces (or steps between them), we can here investigate ways of adsorption on defective positions on nanostructured support with sites formed after a reconstruction in presence of TiCl${}_{4}$.

Starting from model nanoplatelets (50MgCl${}_{2}$ and 50MgCl${}_{2}$/3TiCl${}_{4}$) identified by non-empirical structure prediction based on the genetic algorithm and DFT, as reported in recent papers of us [55,57,58], different ways of binding were proposed and analysed. Vibrational and Raman simulations were carried out to identify whether or not it is possible to identify clear fingerprints of many possible adducts donor/TiCl${}_{4}$ on nano-shaped MgCl${}_{2}$. The possible formation of homogeneous-like TiCl${}_{4}$EB and TiCl${}_{4}$EB${}_{2}$ species was also considered. In the present paper, for simplicity, we address the location of ethyl benzoate (EB) on MgCl${}_{2}$ and MgCl${}_{2}$/TiCl${}_{4}$ precatalyst, even if EB is not so largely adopted in industrial process. For sure, the number of experimental works pointed towards EB [59,60,61,62,63] by far overcomes studies on any other electron donor, that make it easy to compare results with the literature reports, at least in IR case. On the possibility to distinguish between different complexes, the Raman response for sure appears more informative but, at this very moment, experimental data are still not available on these systems even if groups are working on this topic and we foresee a future comparison between simulated and experimental data. Although the present findings are limited to the case of EB, the same approach can be easily applied to industrially more relevant electron donors; we are also immediately going to rehiterate the analysis to ternary systems fully obtained by machine learning techniques that is nanoplatelet of 50MgCl${}_{2}$ reconstructed in presence of both TiCl${}_{4}$ and IDs.

## 2. Computational Models and DFT Calculations Details

As support for the adsorption of EB we adopted models of 50MgCl${}_{2}$ and 50MgCl${}_{2}$/3TiCl${}_{4}$ nanoplates, computed by employing non-empirical structure determination thanks to a software that combines global structure search, based on a genetic algorithm, and local geometry optimization by means of DFT [57,58]. Geometry optimization in first-principles calculations generally seeks a local minimum in the neighborhood of the initial structure, and thus the validity of the obtained structure depends on the initial structure. On the other hand, for complex materials such as Ziegler-Natta catalysts with adsorbates and nanostructures, it is extremely difficult to estimate a reasonable initial structure at an atomic precision. In our previous work, we developed a non-empirical structure determination method for Ziegler-Natta catalysts. It randomly generates initial structures and performs geometry optimization at the DFT level. The energy after the geometry optimization is used as the performance of the corresponding initial structure, and the genetic algorithm evolves the initial structures so as to lower the energy. By repeating the evolution, the method is able to give a reasonable structure without requiring the previous knowledge. The size of nanoclusters is about 3 nm in diameter, comparable with the experimental size of the catalyst primary particles (ca. 2.4–4.0 nm) [40]; at the meantime the ratio between TiCl${}_{4}$ molecules and MgCl${}_{2}$ units equal to 50/3 corresponds to 2.69 wt%, similar to Ti content in a typical Ziegler-Natta catalyst [57,58,64,65]. The DFT [66,67] calculations reported in this paper were mainly based on the B3LYP global hybrid functional [68,69], as implemented in the Crystal program [70,71].

London interactions strongly influence the description of these materials containing not only Ti${}_{x}$Cl${}_{y}$ species but also organic molecules; hence, accounting for long-range correlation, was mandatory. The semi-empirical DFT-D2 approach of Grimme and co-workers [72,73] was adopted together with the hybrid B3LYP functional; it is well assessed that combined procedure may successfully describe surface interactions and adsorption process also including organic molecules [74,75]. Single point energy calculations were run with M06 functional [76] on B3LYPD-2 minima. In fact, the reliability of the semi-empirical DFT-D2 approach as applied to the systems of interest here can be questionable to some extent. The highly parameterized forms of hybrid meta-GGA exchange–correlation functionals, the M06 suite is designed, for application in the area of non-covalent interactions and transition metal bonding. These DF hamiltonians are specifically suited to predict energy estimates for highly correlated systems featuring transition metals and non-bonded interactions. Therefore for a better evaluation of the energetics discussed in this work M06 estimates have been also considered. Split valence triple-zeta basis sets plus polarization (TZVP) of Gaussian type functions have been specifically customized and adopted for all the elements (Mg, Ti and Cl atoms) [49]. Ahlrichs VTZ plus polarization quality [77] have been adopted for the adsorbed organic counterparts. The estimated energies have been corrected for Basis Set Superposition Error (BSSE). To inspect the molecular vibrational shifts of main modes involving CO group, O-C-O and TiCl${}_{4}$ due to both the interaction with the support and the reciprocal interaction of organic and inorganic molecules, we run geometry optimizations followed by harmonic vibrational frequency calculations of the molecular adducts together with the corresponding IR and Raman intensities. Details about the computational set up and the calculations of the vibrational frequencies can be found in previuos works by us [55,78,79].

## 3. Results and Discussion

### 3.1. Models of Ternary Systems MgCl${}_{2}$/TiCl${}_{4}$/EB

The models of coadsorption of EB and TiCl${}_{4}$ on nanoplatelet of MgCl${}_{2}$ may be collected in three groups.

We built a first group of models (A, B, C, J in Figure 1) where the EB is added in different positions with respect to TiCl${}_{4}$ aiming to investigate the effect of reciprocal perturbation on the vibrational response of both internal donor and precursor of active species. Here, due to the heterogeneity of sites, the adsorption of both TiCl${}_{4}$ moieties (and its dimeric form Ti${}_{2}$Cl${}_{8}$) and EB on defective sites has been possible with many combinations of reciprocal positions donor/precatalytic species: edge-like for one between them, EB next to Ti moieties, EB far enough from Ti species to remain more or less unperturbed.

A second group (models D, E, G, H) accounts for the possibility of adsorption of homogeneous-like complexes, already present in reaction medium, to nanoclusters with different Ti/EB ratio, i.e., TiCl${}_{4}$EB and TiCl${}_{4}$(EB)${}_{2}$ on various sites. Previous calculation on internal donor plus TiCl${}_{4}$ adducts have been carried out by some of us on perfect surfaces at different degrees of coverage, in that case we considered tetracoordinated Mg${}^{2}$${}^{+}$ ions of the type exposed on (110) surface [59] and (107) surfaces to adsorb EB in different ways of binding on perfect surfaces.

A third group of models attains the Ti${}_{2}$Cl${}_{8}$EB complexes (F, I) on pentacoordinated Mg${}^{2}$${}^{+}$ sites because historically these type of adducts emulate those formed on (104)-like surfaces claimed to be able to bind Ti${}_{2}$Cl${}_{8}$, that was previously identified as the stereo-selective active species for polypropylene [36]. Although QM calculations proved that adsorption of Ti${}_{2}$Cl${}_{8}$ dimers on MgCl${}_{2}$ surfaces is not feasible [49], the presence of the donors might modify the energy of the adsorption process, giving a new chance for the adduct formation. In the case of adducts EB/Ti${}_{2}$Cl${}_{8}$, after the adsorption on a penta-coordinated row of Mg${}^{2}$${}^{+}$ sites, the dimer breaks in an adsorbed TiCl${}_{4}$EB complex and a TiCl${}_{4}$ molecule physisorbed on the row; after the remotion of physisorbed TiCl${}_{4}$ molecule, the TiCl${}_{4}$EB adsorbed on cluster has been investigared further as Model I. Model F presents a molecule of EB bound to four-coordinated Mg${}^{2}$${}^{+}$ and adjacent to Ti${}_{2}$Cl${}_{8}$ dimer.

For most reasonable models, IR and Raman spectra have been calculated and reported in Figure 2, whereas the energetics associated to each adsorption process is reported in Table 1 (in terms of total electron energy) together with the corresponding reaction.

### 3.2. Adsorption Properties of Adducts on MgCl${}_{2}$ Nanoplatelets

Adsorption energies of selected models are collected in Table 1. Models A, B, C, J represents different coadsorption possibilities for TiCl${}_{4}$ molecules and EB molecules. In the case A, one EB molecule is bound to isolated T site and close to a row of pentacoordinated Mg${}^{2+}$ sites hence reproducing a sort of edge condition. Models B and C similarly reproduces different EB on a P site: the first is perturbed by the presence of TiCl${}_{4}$ whereas the latter is almost unaffected by TiCl${}_{4}$. In all three cases the adsorption energies are close each other (78–79 kJ/mol) with only a tiny larger energy in the adsorption to T site (A) but we can assume that difference falls in the error bar. In model J the co-adsorption involves two EB molecules next to TiCl${}_{4}$, the energy involved in the adsorption of a second molecule on an adjacent site to TiCl${}_{4}$, as indicated in the reaction reported in Table 1, is expected to be highly favored from an enthalpic point of view (Δ*E* = −121.9 kJ/mol). In case F, the perturbation comes from the dimer Ti${}_{2}$Cl${}_{8}$ on sites of P type; the adsorption energy predicted for EB in presence of Ti species strengthens the binding of EB up to 99.3 KJ/mol, in fact in this case a chelation through O(R) occurs resembling a diester mode coordination. The other models D, G, E, H, I concern the adsorption of TiCl${}_{4}$EB${}_{x}$ (x = 1, 2) complexes, on defective T positions (D, G, E, H) and on P position close to a couple of T Mg${}^{2+}$ sites (I). The mono-EB specie is weakly bound to T sites whereas the largest binding energies belong to the di-EB homogeneous-like adducts. A stronger interaction is predicted in the case G respect to D due to the presence of an adjacent most acidic T site. In the case G, the complex TiCl${}_{4}$EB is bound to one of a couple of adjacent T sites resembling 110-like sequence; that differs from the outcome of previous periodic 2D calculations where the structure TiCl${}_{4}$EB/(110) evolves towards a new one with EB directly bound to support by carbonyl oxygen and at the meantime to a five coordinated TiCl${}_{4}$ (Mg-O-Ti binding) [59]. This can be rationalized in terms of donor-donor interaction when periodic boundary conditions are assumed (even at low degree of coverage) and interactions of donors with adjacent layers of MgCl${}_{2}$. In the models presented here, bulky donors have larger degrees of freedom to keep a Mg-Ti-O binding. TiCl${}_{4}$ in TiCl${}_{4}$EB is partially detached from surface and that leaves coordination vacancies around titanium where a further EB molecule can easily insert to get a six-fold coordination to Ti and then a significant gain in energy (models E, H). In model I the adsorption energy of TiCl${}_{4}$EB on P sites is close to the corresponding energy on T sites (model G), supporting the hypothesis that the presence of donor allows the binding of TiCl${}_{4}$ on rows of (104)-like rows in contrast to a not effective binding of free TiCl${}_{4}$ on (104) surfaces as predicted in literature. Definitely, it seems that in models close to truly nanosized structures, the clean break between the adsorption behaviour of T and P sites, emerging from a periodic approach, becomes smoother; that can be due to the proximity of a few T sites to other P centers with a consequent unexpected larger polarizing effect by P sites respect to surface case. The adsorption energy reveals that the binding of EB is only tiny sensitive to the site of adsorption due to the frontier behaviour of sites. In particular, in the case of model B, where the proximity of TiCl${}_{4}$ increases the polarizing ability of pentacoordinated Mg${}^{2}$${}^{+}$ (as we already found in probing cluster sites by CO [55]), the adsorption energy almost equals that of model A. In the case of homogeneous-like complexes, the complex with two EB molecules (model E) is by far more strongly bound to the support, in fact in mono-EB case (models D, G), a large distortion occurs with respect to the geometry adopted from molecule of EB in the gas phase with a cost amounting to 34.5 kJ/mol. For model D, the significant difference with the results obtained for extended surfaces [59] is due to the fact that the minimum is unstable; it evolves towards a structure where EB is bound to an adjacent tetracoordinated site: in other words the TiCl${}_{4}$EB is bonded to MgCl${}_{2}$ through the oxygen of the ester group.

Table 1 shows that the differences in M06 and B3LYP-D2 adsorption energies are negligible with the only exceptions for D, G, I minima i.e., the adsorption processes involving the TiCl${}_{4}$EB complex where M06 estimates are significantly larger. That may be attributed to the lower cost predicted, at M06 level, for the deformation of the complex on cluster respect to the molecule in gas phase.

Mulliken analysis has been performed on selected models. In particular, we considered how Mulliken charges of Ti and of the two Cl atoms bound exclusively to Ti (free chlorine atoms exposed to reaction medium) of adsorbed MgCl${}_{2}$/TiCl${}_{4}$ are modified in the presence of one or two co-adorbed EBs as in models B and J, respectively. In model B we find the following charges: Ti (+1.567 |e|), Cl (−0.362 and −0.286 |e|), whereas in model J we got: Ti (+1.579 |e|), Cl (−0.386 and −0.286 |e|); corresponding values for MgCl${}_{2}$/TiCl${}_{4}$ are for Ti (+1.547 |e|) and for Cl (−0.268 and 0.286 |e|). Even between the limits of a Mulliken treatment, we can infer that the presence of EB (1 or 2 molecules) increases the charge on Ti and also the difference between the charges on the two Cl atoms bound to it, thus enhancing the not equivalency of the two Ti-Cl bonds that could be relevant in the catalytic process starting with the cleavage of one Ti-Cl bond. Nevertheless, this suggestion should be more properly investigated by considering the electronic effect of donors in presence of the reduced form of Ti i.e., (Ti(III)) that is supposed to be the active species in Ziegler-Natta polymerization catalysis [80,81]. The effect of the presence of donors on charge density, charge transfer/redistribution and on catalysis is a topic that we are investigating with different methodologies but is out of the aims of the present paper.

### 3.3. IR and Raman Response of EB/TiCl${}_{4}$ on Nanoplatelet

#### 3.3.1. IR Simulations

Panel a of Figure 1 reports on left side the IR spectrum of MgCl${}_{2}$/3TiCl${}_{4}$ system in low frequency range 400–550 cm${}^{-1}$ whereas on the right side of the same panel it is reported the spectrum of EB molecule.

We first recap main IR feautures of EB molecule. The spectrum is overlooked by two intense peaks at 1766 and 1277 cm${}^{-1}$; the band at 1766 cm${}^{-1}$ is assigned to ν(C=O) stretching mode, whereas that at 1277 cm${}^{-1}$, coupled with the bands at 1130 and 1140 cm${}^{-1}$, is due to a ν(C-O) stretching mode in the (C-O-C) group. Most of the other weak bands in the spectrum are related to C-H stretching (about 3000 cm${}^{-1}$) or deformation modes of the phenyl ring and of the ethyl group, (e.g., the weak bands at 1200 and 1098 cm${}^{-1}$ is associated with -CH bending of the phenyl ring whereas 1334, 1349 cm${}^{-1}$ modes are due to ring deformation and the very weak bands at 1403 and 1432 cm${}^{-1}$ are associated to the wagging of CH${}_{3}$). It is very widely reported that the positions of the absorption bands ν(C-O) and ν(C-O-C) change upon EB complexation, whereas the others are much less sensitive.

Due to the fact that we investigate a coadsorption with Ti${}_{x}$Cl${}_{y}$, also the region 400–550 cm${}^{-1}$ has been here reported since our recent investigations on binary systems TiCl${}_{4}$/MgCl${}_{2}$ revealed that the region cannot be neglected in IR spectra and even more in Raman analysis.

Concerning the IR response of MgCl${}_{2}$/Ti*x*Cl${}_{y}$ nanoplatelets, it covers a region between 200 and 500 cm${}^{-1}$ [55]; altough the 200–400 cm${}^{-1}$ region refers to bulk modes, strongly dependent on the particle shape and size, the 400–500 cm${}^{-1}$ region contains fingerprints that can be easily correlated to surface sites and adorbed species: peaks at 429 and 445 cm${}^{-1}$ related to exposed tetracoordinated Mg${}^{2+}$ typical of 110 surfaces, a couple of bands at 465 and 485 cm${}^{-1}$ attributed to symmetric and antisymmetric stretching of Ti-Cl bonds in supported TiCl${}_{4}$ and 458, 476 and 495 cm${}^{-1}$ bands for supported Ti${}_{2}$Cl${}_{8}$.

We considered now the adsorption of EB, according to the models sketched in Figure 1. IR spectra are reported in Figure 2, by considering separately the two vibrational regions 400–550 cm${}^{-1}$ and 1200–1800 cm${}^{-1}$ in order to better highlight the Ti${}_{x}$Cl${}_{y}$ and the EB response. Panels b, c and d of Figure 2 refer to model A-C that corresponds to EB coadsorption at almost isolated tetracoordinated and pentacoordinated Mg${}^{2+}$ (panels b and d) and EB coadsorbed at a pentacoordinated Mg${}^{2+}$ in proximity of a TiCl${}_{4}$ surface specie of the precatalist (panel c). Panels e and f of Figure 2 refer to models D–E of Figure 1. Panel g refers to model F. In Model D and F the TiCl${}_{4}$/EB and TiCl${}_{4}$/(EB)${}_{2}$ homogeneous-like complexes are loosely bound to the MgCl${}_{2}$ surface through bridge chlorine atoms.

The main features of the spectra are:the ν(C=O) vibration that dominates the IR spectrum and is located at 1683, 1697 and 1695 cm${}^{-1}$ for models A–C, downshifted by 83, 69 and 71 cm${}^{-1}$ in reference to gas phase, respectively. Two very weak satellite bands are also osberved at 1638 and 1615 cm${}^{-1}$ that correspond to C-C stretching modes of the phenyl ring. In the presence of a TiCl${}_{4}$EB adduct (model D) the C=O stretching mode undergoes a huge redshift to 1587 cm${}^{-1}$ (medium); it is further redshifted to 1556 cm${}^{-1}$ and coupled with a very weak band at 1585 cm${}^{-1}$ in the case of adduct E: they correspond to the antisymmetric and to the symmetric coupling of the two C=O stretchings;the ν(C-O) signal drops its intensity with respect to free EB and is splitted into three components at 1308, 1323, 1336; 1312, 1332, 1348 and 1313, 1329, 1344 cm${}^{-1}$ for models A–C, respectively due to coupling with CH${}_{2}$ twisting and phenyl H modes; however for model A the triplet of bands have comparable intensity, whereas for model B and C the band at higher wavenumber dominates in the triplet. A similar group of bands is observed for models D and E with the most intense peak located at 1362 cm${}^{-1}$;a new weak band appears at 1414, 1418, 1417 cm${}^{-1}$ for models A–C, and at 1427 cm${}^{-1}$ for models D–E, is associated to the wagging of -CH${}_{3}$ and therefore slightly perturbed by different EB binding mode; the band is weak but it can be clearly identified in the IR spectra;in the spectral region 400–500 cm${}^{-1}$ region, for A and B models the Ti-Cl stretching mode at 465 and 485 cm${}^{-1}$ are only slightly perturbed by the presence of EB; for model C, the coupling with EB mode causes the splitting of Ti-Cl symmetric stretching mode in components at 464, 465, 468 cm${}^{-1}$ and antisymmetric Ti-Cl stretching at 483, 484, also partially coupled with Mg-Cl modes of tetrahedral Mg${}^{2+}$ at 490 cm${}^{-1}$. For models D–E two further bands, not visible in Raman, pop-up at 437 cm${}^{-1}$ and at 449 cm${}^{-1}$ due to bending modes of O-Ti-O and Ti-O-C, respectively. Weak bands at 588, 645 cm${}^{-1}$ assigned to Ti-O-C appear.

The adopted models (50MgCl${}_{2}$ and 50MgCl${}_{2}$/3TiCl${}_{4}$) whose size is about 3 nm in diameter allow the occurrence of sites (and couple of sites) where different possible ways of binding for EB are taken into account also in relation to Ti${}_{x}$Cl${}_{y}$ species. Thus, from that size on, the internal donor’s effect and the fingerprint we discussed cannot be significantly influenced by the shape or dimensions of the nanocluster. Hence the larger effect of those variables on spectroscopic response can be assumed to be limited to the bulk modes in the region of 200–400 cm${}^{-1}$ in IR spectra, as shown in our previous report [55].

#### 3.3.2. Raman Characterization MgCl${}_{2}$/TiCl${}_{4}$/EB Adducts

Let us consider first the EB molecule by itself in gas phase, see panel a of Figure 2. The main differences with respect to the IR counterpart are: (i) the much higher intensity of the 1640 and 1620 cm${}^{-1}$ bands, due to C-C stretching of the phenyl ring; (ii) a band falling at 1019 cm${}^{-1}$ related to phenyl ring deformations; (iii) a significant increase in intensity for the triplet 1483, 1489, 1501 cm${}^{-1}$ due to modes involving the O-C-O group.

For EB supported adducts, Raman spectra differ from IR spectra in the presence of the two intense bands at ∼1636 and 1612 cm${}^{-1}$ related to the C-C stretching of the phenyl ring that appear almost unperturbed in all the models; in the model E, at variance of IR, the antisymmetric and symmetric coupling of the two C=O stretchings give rise to peaks of comparable intensity at 1556 and 1585 cm${}^{-1}$, thus unambiguously probing the presence of TiCl${}_{4}$(EB)${}_{2}$.

All the other peaks, even if with different relative intensity, are common to the IR and Raman spectra.

### 3.4. Discussion

EB is adsorbed at T and P isolated sites (models A and C) and at T and P sites adjacent to preadsorbed TiCl${}_{4}$ and Ti${}_{2}$Cl${}_{8}$ (models B and F) giving rise to surface adducts of comparable stability (∼80 kJ/mol, B3LYP-D2 and M06 results, Table 1). Once adsorbed to support Ti${}_{2}$Cl${}_{8}$EB cannot be identified since adsorption causes the fragmentation of the Ti cluster into TiCl${}_{4}$EB (model I) and physisorbed TiCl${}_{4}$. The formation of the MgCl${}_{2}$/EB adducts is monitored by the IR spectra.

Line a of Figure 3 reports a convolution of IR spectra of the most relevant MgCl${}_{2}$/EB surface structures in presence of TiCl${}_{4}$ (sum of simulated intensities for models A, B, C and F). Inspection of the spectrum shows that the two main features are the ν(C=O) peak, sharp and narrow (its width is ∼12 cm${}^{-1}$), centered at 1678 cm${}^{-1}$ and the ν(C-O-C) vibration that leads to a group of signals in the 1275–1350 cm${}^{-1}$ range; ν(C=O) peak appears to be sensitive to the coordination number of Mg${}^{+2}$, but scarcely affected by the chemical environment of the surface cations; as a consequence, the shape of the ν(C-O) signal can unambiguously probe a simultaneous presence of tetra and pentacoordinated Mg${}^{+2}$ sites (compare panel d to panel f of Figure 2).

The homogeneous like TiCl${}_{4}$EB and TiCl${}_{4}$EB${}_{2}$ complexes can easily bind at the the Mg${}^{+2}$ sites of MgCl${}_{2}$. However the only species present on P sites of the (104)-like row is the TiCl${}_{4}$EB (model I) in contrast, on T sites (i.e., (110)-like sites) both TiCl${}_{4}$EB and TiCl${}_{4}$EB${}_{2}$ can be coordinated. The relative binding energies for TiCl${}_{4}$EB adducts are quite similar to previously discussed models, ΔE∼70–80 kJ/mol (B3LYP-D2), but much larger for TiCl${}_{4}$EB${}_{2}$, ΔE∼170–180 kJ/mol (B3LYP-D2).

Convoluted IR spectrum of adsorbed TiCl${}_{4}$EB${}_{x}$ complexes (sum on spectra of models D and E) reported as line b of Figure 3, show that the TiCl${}_{4}$EB complex is associated with two signals centered at about 1620 cm${}^{-1}$ due to ring deformations coupled to C=O stretching whereas, TiCl${}_{4}$/(EB)${}_{2}$ is unambiguously characterized by a band at 1556 cm${}^{-1}$, due to carbonyl couples, that dominates the IR spectrum.

We can notice the excellent agreement between computed IR spectra of Figure 3a,b and the experimental one of MgCl${}_{2}$/EB/TiCl${}_{4}$ from Ref. [59] and reported in Figure 3c; the larger discrepancy can be found in the region of ν(C=O) (CO stretching) with a shift that can be easily attributed to anharmonicity of the mode, neglected in the harmonic approximation we adopted. Concerning the Raman response, convoluted spectra are reported in Figure 3d,e. The high frequency region is dominated by two intense peaks at 1720 cm${}^{-1}$ due to ν(C=O) and at 1660 cm${}^{-1}$ associated to C-C stretching mode of the phenyl ring; the 1275–1350 cm${}^{-1}$ range shows a group of signals related to ν(C-O). Adsorbed TiCl${}_{4}$EB${}_{1,2}$ provide clear fingerprints in the Raman spectra: the couple of bands at 1636 (with a shoulder at 1621 cm${}^{-1}$) and at 1587 cm${}^{-1}$ identifies the mono-EB adduct, whereas the presence of an additional peak at 1556 cm${}^{-1}$ reveals the presence of the di-EB adduct.

The formation of homogeneous—like TiCl${}_{4}$EB${}_{1,2}$ complexes at the MgCl${}_{2}$ nanoplatelets is thus unambiguously probed by IR and Raman spectra. However we can wonder about the role, if any, of the MgCl${}_{2}$ substrate in promoting the formation of those surface complexes. Starting from MgCl${}_{2}$/EB, TiCl${}_{4}$ can be bound on a couple of adjacent T sites even in the presence of preadsorbed EB at one of the T centers (MgCl${}_{2}$/TiCl${}_{4}$…EB, model B of Figure 1 and Table 1) with a binding energy of −91.6 kJ/mol at M06 approximation, (the corresponding value on bare MgCl${}_{2}$ is −69.0 kJ/mol [55]). In the same way, starting from MgCl${}_{2}$/TiCl${}_{4}$, EB can absorb at the one of the T sites that binds TiCl${}_{4}$ (again model B) with a coordination energy of −106.9 kJ/mol (M06). Despite sharing the same Mg${}^{+2}$ center, both TiCl${}_{4}$ and EB are strongly bound to the substrate; thus a surface reaction starting from MgCl${}_{2}$/TiCl${}_{4}$…EB (Model B) and leading to MgCl${}_{2}$/TiCl${}_{4}$(EB) (model G) is energetically not favored by ∼61 kJ/mol, Table 1. We consider now a couple of EBs adsorbed on adjacent T sites; coordination of TiCl${}_{4}$ involving the same T sites is again energetically favored by −103 kJ/mol: hence, coadsorption, first with one and then with two EBs progressively improves TiCl${}_{4}$ coordination to MgCl${}_{2}$. The second EB molecule succesfully coadsorbes on MgCl${}_{2}$/TiCl${}_{4}$EB leading to Model J with an associated energy variation ΔE = −123.6 kJ/mol. Hence, EB and TiCl${}_{4}$ act cooperatively and stabilize each other so that, the presence of a second coadsorbed EB further disfavors a spontaneous evolution to the homogeneous like TiCl${}_{4}$(EB)${}_{2}$ surface adduct (Model H) and the corresponding reaction is disadvantaged by 68 kJ/mol, Table 1. Similarly to what happened in the case of adsorption processes we found that a surface reaction starting from MgCl${}_{2}$/TiCl${}_{4}$…EB (Model B) and leading to MgCl${}_{2}$/TiCl${}_{4}$ (EB) (model G) is energetically not favored by 61 kJ/mol in M06 case against 35 kJ/mol predicted by Grimme method; whereas the evolution to the homogeneous like TiCl${}_{4}$(EB)${}_{2}$ surface adduct is disfavored by 68 kJ/mol at M06 level against 55 kJ/mol in B3LYP-D2 case. Indeed, it is experimentally known that TiCl${}_{4}$/EB complexes dissociatively adsorb on MgCl${}_{2}$ [35].

Finally, it is worth mentioning that the presence of donor allows the binding of TiCl${}_{4}$ on rows of 104-like Mg${}^{2+}$ sites (model I) contrary to an effective no binding of single TiCl${}_{4}$ predicted in literature on (104) surfaces.

### 3.5. Conclusions

Ternary MgCl${}_{2}$/TiCl${}_{4}$/EB systems working as precatalyst in olefin polymerization are here presented as paradigmatic complex materials with adsorbates and nanostructures; here in addition to the difficulty to estimate a reasonable structure of the support, the additional issue of coadsorption of internal donors (ethylbenzoate) and Ti${}_{x}$Cl${}_{y}$ on support have been addressed, too. As support, we adopted models of 50MgCl${}_{2}$ and 50MgCl${}_{2}$/3TiCl${}_{4}$ nanoplates, computed by employing non-empirical structure determination thanks to a software that combines global structure search, based on a genetic algorithm, and local geometry optimization by means of DFT. Different ways of binding of EB, also in relation to adsorbed TiCl${}_{4}$ or Ti${}_{2}$Cl${}_{8}$, have been modelled together with their spectroscopic IR and Raman response.

Unlike surface predictions, a smooth difference exist in adsorption energy on tetracoordinated or pentacoordinated Mg${}^{+2}$ sites due to the close proximity of different sites with different acidity. To confirm the energetics of adsorption together with B3LYP-D2 calculations, M06 calculations were performed and the two sets of data turn out to be in farewell agreement, meta-GGA DF providing slightly larger values.

Inspection of the IR spectrum shows that the two main features are the ν(C=O) peak, sharp and narrow (its width is ∼12 cm${}^{-1}$), centered at 1678 cm${}^{-1}$ and the ν(C-O-C) vibration that leads to a group of signals in the 1275–1350 cm${}^{-1}$ range. ν(C=O) peak appears to be sensitive to the coordination number of Mg${}^{+2}$, but scarcely affected by the chemical environment of the surface cations; therefore, the characteristic shape of the ν(C-O) signal can unambiguously probe a simultaneous presence of tetra and pentacoordinated Mg${}^{+2}$ sites (compare panel d to panel f of Figure 2). Computed IR spectra are in excellent agreement with available experimental data.

The homogeneous like TiCl${}_{4}$EB and TiCl${}_{4}$EB${}_{2}$ complexes can easily bind at the the Mg${}^{+2}$ sites of MgCl${}_{2}$. However the only species present on P sites of the (104)-like row is the TiCl${}_{4}$EB; on T sites (i.e., (110)-like sites) both TiCl${}_{4}$EB and TiCl${}_{4}$EB${}_{2}$ can be coordinated. The relative binding energies are quite similar for TiCl${}_{4}$EB adducts, ΔE ∼ 70–80 kJ/mol (B3LYP-D2), but much larger for TiCl${}_{4}$EB${}_{2}$, ΔE ∼ 170–180 kJ/mol (B3LYP-D2). Adsorbed TiCl${}_{4}$EB${}_{1,2}$ provide clear fingerprints both in the IR and Raman spectra: the couple of bands at 1636 (with a shoulder at 1621 cm${}^{-1}$) and at 1587 cm${}^{-1}$ identifies the mono-EB adduct, whereas the presence of an additional peak at 1556 cm${}^{-1}$ undoubtedly identifies the presence of the di-EB adduct. However energetics indicates that TiCl${}_{4}$/EB complexes preferentially adsorb on MgCl${}_{2}$ in a dissociative way, in agreement with observations. In addition, the presence of donor permits the binding of TiCl${}_{4}$ on rows of (104)-like sites (model I) in contrast to what is reported in literature for regular surfaces.

## Figures and Tables

**Figure 1 materials-15-00909-f001:**
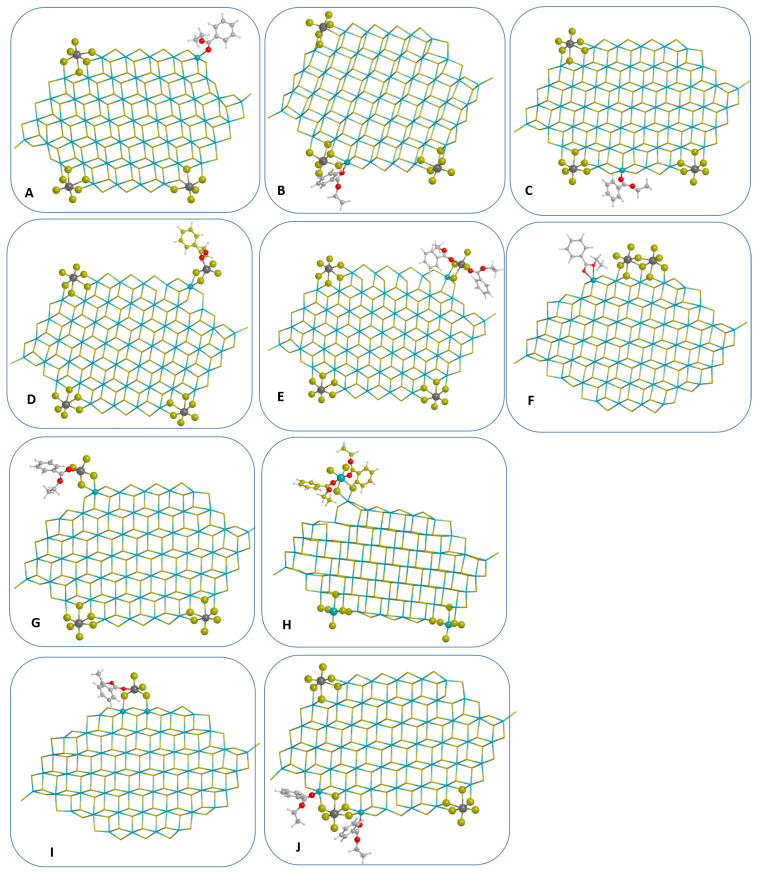
Coadsorption models of nanoplatelets (50MgCl${}_{2}$/TiCl${}_{4}$, see text for details) and EB with different way of binding (**A**–**C**); models of homogeneous-like TiCl${}_{4}$ (EB)${}_{x}$ (x = 1, 2) complexes on the same platelet (**D**,**E**,**G**,**H**) and models obtained after the coadsorption of EB and Ti${}_{2}$Cl${}_{8}$ on naked MgCl${}_{2}$ nanoplatelet (**F**,**I**); co-adsorption model involving two EB molecules next to TiCl${}_{4}$ (**J**). Optimizations have been performed at B3LYP-D2/TZVP level. Chlorine, Magnesium and Titanium atoms are represented in green, yellow and dark grey, respectively. All generic atoms belonging to the nanoplatelet are represented as sticks, whereas atoms of TiCl${}_{4}$ and EB molecules and Mg atoms involved in the adsorption process are reported as balls.

**Figure 2 materials-15-00909-f002:**
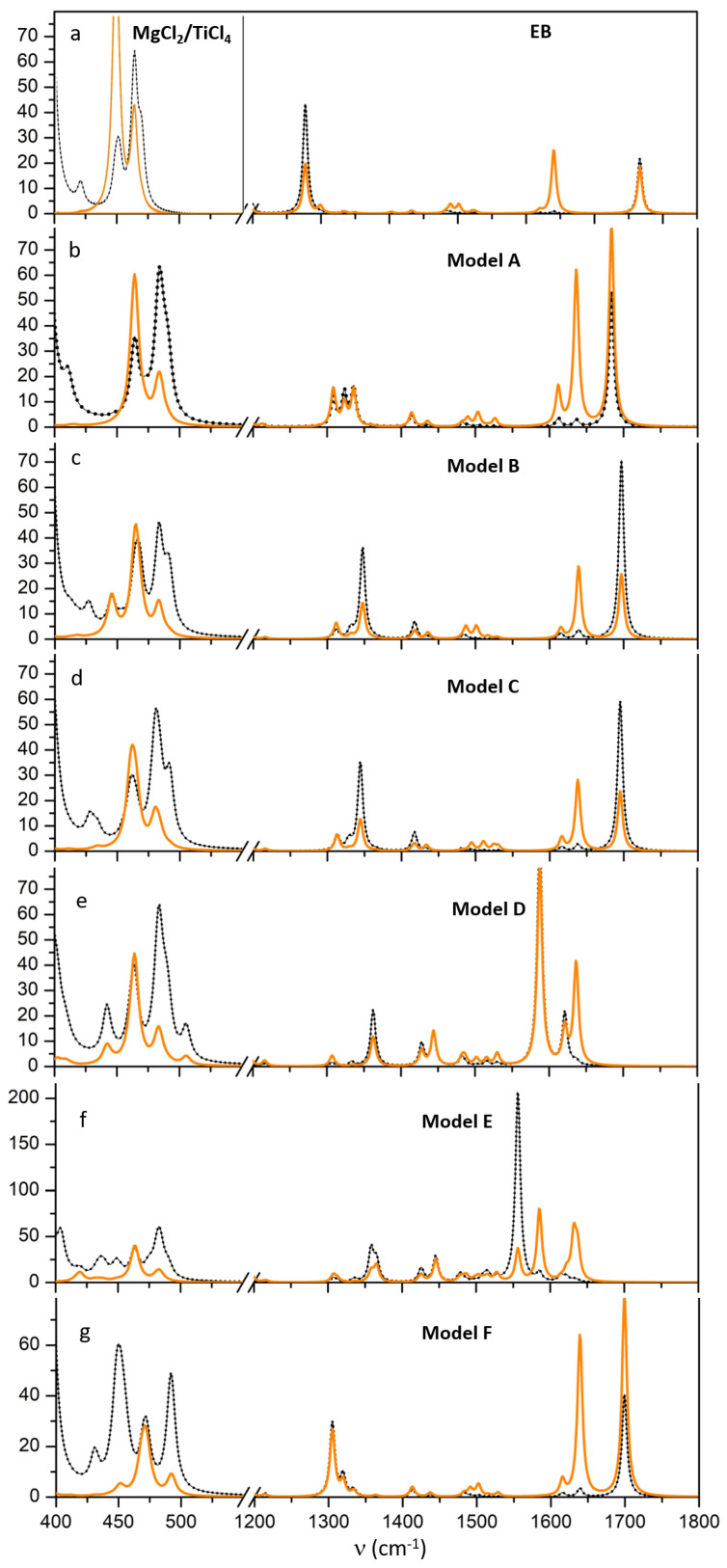
Simulated IR (black, dotted lines) and Raman spectra (orange, solid lines) at B3LYP-D2/TZVP for models A–F proposed in Figure 1, in the spectral regions 400–550 cm${}^{-}$${}^{1}$ and 1200–1800 cm${}^{-}$${}^{1}$ (panels **b**–**g**) and for 50MgCl${}_{2}$/3TiCl${}_{4}$ and EB molecule in the regions of interest (panel **a**). Intensity are in (Km/mol) and arbitrary units, respectively.

**Figure 3 materials-15-00909-f003:**
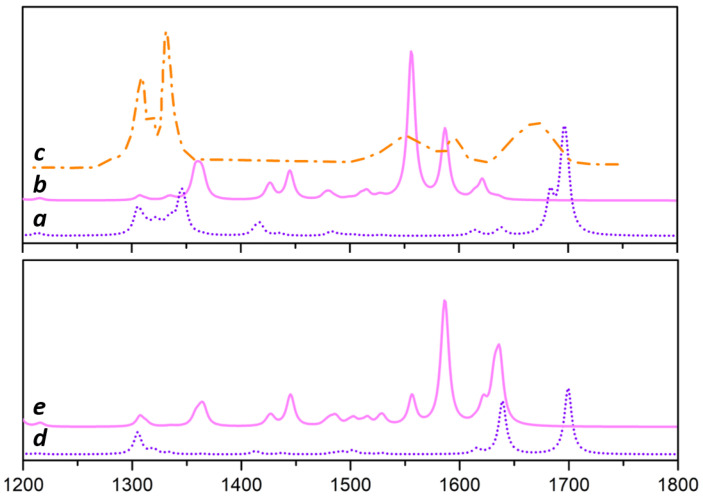
Sum of simulated IR intensity at B3LYP-D2/TZVP, for co-adsorption models A, B, C, F (line a) and for homogeneous-like adducts, models D, E (line b). Experimental IR spectrum from Ref. [59] reported in the 1200–1350 and 1550–1800 cm${}^{-1}$ spectral range (line c). Sum of simulated Raman intensity at B3LYP-D2/TZVP, for co-adsorption models A, B, C, F (line d) and for homogeneous-like adducts, models D, E (line e). The spectral region considered is 1200–1800 cm${}^{-1}$, the intensities are in arbitrary units.

**Table 1 materials-15-00909-t001:** Energetics of adsorption of EB, TiCl${}_{4}$EB and TiCl${}_{4}$EB${}_{2}$ complexes for models reported in Figure 1 (energies in kJ/mol). All calculations were performed by adoption of B3LYP-D2 and M06 functionals and Ahlrichs VTZP basis set. In M06 case, single point energy calculations have been computed on geometries optimized at B3LYP-D2 level. All data are BSSE corrected.

Reaction	Model	Δ*E* (B3LYP-D2)	Δ*E* (M06)
EB${}_{gas}$ + (MgCl${}_{2}$TiCl${}_{4}$)${}_{surf}$ → (MgCl${}_{2}$TiCl${}_{4}...$EB)${}_{surf}$	A	−79.3	−81.6
“	B	−78.9	−80.9
”	C	−77.5	−79.5
TiCl${}_{4}$EB${}_{gas}$ + (MgCl${}_{2}$TiCl${}_{4}$)${}_{surf}$ → (MgCl${}_{2}$TiCl${}_{4}$EB)${}_{surf}$	D	−66.3	−75.4
TiCl${}_{4}$EB${}_{2}$${}_{gas}$ + (MgCl${}_{2}$TiCl${}_{4}$)${}_{surf}$ → (MgCl${}_{2}$TiCl${}_{4}$EB${}_{2}$)${}_{surf}$	E	−181.2	−185.8
EB${}_{gas}$ + (MgCl${}_{2}$Ti${}_{2}$Cl${}_{8}$)${}_{surf}$ → ((MgCl${}_{2}$Ti${}_{2}$Cl${}_{8}...$EB)${}_{surf}$	F	−99.3	−96.0
TiCl${}_{4}$EB${}_{gas}$ + (MgCl${}_{2}$TiCl${}_{4}$)${}_{surf}$ → (MgCl${}_{2}$TiCl${}_{4}$EB)${}_{surf}$	G	−79.3	−94.9
TiCl${}_{4}$EB${}_{2}$${}_{gas}$ + (MgCl${}_{2}$TiCl${}_{4}$)${}_{surf}$ → (MgCl${}_{2}$TiCl${}_{4}$EB${}_{2}$)${}_{surf}$	H	−172.1	−191.6
TiCl${}_{4}$EB${}_{gas}$ + (MgCl${}_{2}$Ti${}_{2}$Cl${}_{8}$)${}_{surf}$ → (MgCl${}_{2}$TiCl${}_{4}$EB)${}_{surf}$	I	−79.5	−95.3
EB${}_{gas}$ + (MgCl${}_{2}$TiCl${}_{4}...$EB)${}_{surf}$ → (MgCl${}_{2}$TiCl${}_{4}...$EB${}_{2}$)${}_{surf}$	J	−121.9	−123.6
(MgCl${}_{2}$TiCl${}_{4}...$EB)${}_{surf}$ → (MgCl${}_{2}$TiCl${}_{4}$EB)${}_{surf}$	G	35.5	60.9
(MgCl${}_{2}$TiCl${}_{4}...$EB${}_{2}$)${}_{surf}$ → (MgCl${}_{2}$TiCl${}_{4}$EB${}_{2}$)${}_{surf}$	H	55.5	67.8

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
