# Peer review of "Effect of Internal Donors on Raman and IR Spectroscopic Fingerprints of MgCl2/TiCl4 Nanoclusters Determined by Machine Learning and DFT"

_materials, 2022, doi:10.3390/ma15030909_

Round 1
Reviewer 1 Report
In the manuscript, the author presents a theoretical investigation on the effect of internal donors on Raman and IR spectroscopic fingerprints of MgCl2/TiCl4 Nanoclusters. Understanding the origin of MgCl2-based Ziegler–Natta catalyst is an area under extensive research. However, despite this potential interest, the present paper should be improved to address the comments below.
- In the Introduction section, the literature review of MgCl2/TiCl4 simulation is excessively long given the paper's focus on the effect of internal donors.
- The manuscript should describe the machine learning method used in the simulation and the reason it is used in addition to the current writing: "a program that combines global structure search using a genetic algorithm and local geometry optimization using DFT".
- The writing needs to be extensively improved: it has many grammatical mistakes and sentences which are difficult to be understood, and many paragraph structure is not well organized.
- The author should discuss how the fingerprint and the internal donor's effect on it change with respect to nanocluster size and shape.
Following the above considerations, I cannot recommend the present manuscript for publication in Materials in the present form. However, after a reorganization of the contents and fixing the issues, the manuscript can be reconsidered to be published.
Author Response
We thank the Reviewer 1 for his comments and useful suggestions received by our manuscript.
In the revised version, we have responded to the concerns raised by the reviewer. In the following, we report detailed point-by-point reply to your comments for your convenience, the following color code has been used: blue and italic: comments of the reviewer; black: our replies to the reviewer’s comments; red: modifications that were applied in the text to address the indications of the reviewer.
We think that this new amended version can be suitable for publication in Materials.
Torino, 17th January 2022.
Yours sincerely,
Maddalena D’Amore (on behalf of all the authors).
Remarks from the Reviewer
Reviewer: 1
1) Comment – In the manuscript, the author presents a theoretical investigation on the effect of internal donors on Raman and IR spectroscopic fingerprints of MgCl2/TiCl4 Nanoclusters. Understanding the origin of MgCl2-based Ziegler–Natta catalyst is an area under extensive research. However, despite this potential interest, the present paper should be improved to address the comments below. In the Introduction section, the literature review of MgCl2/TiCl4 simulation is excessively long given the paper's focus on the effect of internal donors.
Author reply – We are grateful to the reviewer for finding our paper of potential interest and we think we largely improved the paper, in particular we shortened the discussion concerning the simulation of MgCl2/TiCl4 and we left the part pertaining to donors.
2) Comment – The manuscript should describe the machine learning method used in the simulation and the reason it is used in addition to the current writing: "a program that combines global structure search using a genetic algorithm and local geometry optimization using DFT".
Author reply – We appreciated the consideration expressed by the reviewer that allows us to better describe the methodology we developed and validated.
Author action – In Computational Models and DFT Calculations Details at line 129 and subsequent ones, we introduced the following modifications:
" As support for the adsorption of EB we adopted models of 50MgCl2 and 50MgCl2/3TiCl4 nanoplates, computed by employing non-empirical structure determination thanks to a software that combines global structure search, based on a genetic algorithm, and local geometry optimization by means of DFT Geometry optimization in first-principles calculations generally seeks a local minimum in the neighbourhood of the initial structure, and thus the validity of the obtained structure depends on the initial structure. On the other hand, for complex materials such as Ziegler-Natta catalysts with adsorbates and nanostructures, it is extremely difficult to estimate a reasonable initial structure at an atomic precision. In our previous work, we developed a non-empirical structure determination method for Ziegler-Natta catalysts. It randomly generates initial structures and performs geometry optimization at the DFT level. The energy after the geometry optimization is used as the performance of the corresponding initial structure, and the genetic algorithm evolves the initial structures so as to lower the energy. By repeating the evolution, the method is able to give a reasonable structure without requiring the previous knowledge."
3) Comment – The writing needs to be extensively improved: it has many grammatical mistakes and sentences which are difficult to be understood, and many paragraph structure is not well organized.
Author reply – We improved the writing and amended the text from grammatical mistakes, in addition some sentences have been made more easily readable. In some cases paragraph structure has been modified.
Author action – The subsection: “Adsorption properties of adducts on MgCl2 nanoplatelets“ has been shifted along the manuscript and introduced after the subsection “Models of ternary systems MgCl2/TiCl4/EB” to better organize the paper.
4) Comment – The author should discuss how the fingerprint and the internal donor's effect on it change with respect to nanocluster size and shape.
Author reply –In the IR spectrum we can clearly identify a low wave-number region in the range 235-360 cm-1 that comes from bulk modes. Although this spectral region appears strongly dependent on the shape of the nanoplate, on the number and type of edge ions some common features can be recognized. The band at 429 cm-1 associated to isolated T sites remains unperturbed in model B but is absent in model A where the site T is covered by EB. In subsection 3.2.1 at lines 279-285 of the draft it was already present a quick comment on the effect of cluster size and shape “Concerning the IR response of MgCl2/TixCly nanoplatelets, it covers a region between 200 and 500 cm-1 [57]; altough the 200-400 cm-1 region refers to bulk modes, strongly dependent on the particle shape and size, the 400-500 cm-1 region contains fingerprints that can be easily correlated to surface sites and adorbed species: peaks at 429 and 445 cm-1 related to exposed tetracoordinated Mg2+ typical of 110 surfaces, a couple of bands at 465 and 485 cm-1 attributed to symmetric and antysimmetric stretching of Ti-Cl bonds in supported TiCl4 and 458, 476 and 495 cm-1 bands for supported Ti2Cl8”. We better explain the topic to remove any doubt.
Author action – In Subsection 3.3.1 we introduced the following sentences at lines 344-351:
"The adopted models (50MgCl2 and 50MgCl2/3TiCl4) whose size is about 3 nm in diameter allow the occurrence of sites (and couple of sites) where different possible ways of binding for EB are taken into account also in relation to TixCly species. Therefore, from that size on, the effects of internal donors and the fingerprint we discussed cannot be significantly influenced by the shape or dimensions of the nanocluster. Hence the larger effect of those variables on spectroscopic response can be assumed to be limited to the region of 200-400 cm-1 in IR spectra as shown in our previous report [57]”.
Reviewer 2 Report
The authors present a very clear and detailed article on the current problem in catalysis, as an another step in their ongoing study (Refs. 50-61). The literature review and a description of the problem are presented in great detail and deeply acquaint the reader with the topic. I have only few minor comments/questions:
- What is the point of using M06 XC for single-point calculations? The numbers appear in Table 1, but there is no any explanation or comparison to B3LYP-D2. At least short comment would be great.
- You show that absorption leads to various changes in vibrational spectra, and also assign those modes to specific atoms vibrations. But the explanation for this changes is very brief. I wonder, why don't you analyze charge redistribution/transfer for adsorbates as it is crucial for catalysis?
- E, H, F and L models in Table 1 have lowest energies, but they are incomparable to other models due to different number of adsorbates, as well you describe in the text where the excess of energy comes from for EB2, for example. Maybe it is possible to put energies per adsorbed EB into Table 1 or find another way to present the results of thermodynamics?
- Multiple times in the text you use surface indices without brackets, it can be confusing to the reader.
Author Response
We thank the Reviewer 2 for his comments and useful suggestions received by our manuscript.
In the revised version, we have responded to the concerns raised by the reviewer as outlined. In the following, we report detailed point-by-point reply to your comments for your convenience, the following colour code has been used: blue and italic: comments of the reviewer; black: our replies to the reviewer's comments; red: modifications that were applied in the text to address the indications of the reviewer.
We think that this new amended version can be suitable for publication in Materials.
Torino, 17th January 2022.
Yours sincerely,
Maddalena D’Amore (on behalf of all the authors).
Reviewer:2
1) Comment – The authors present a very clear and detailed article on the current problem in catalysis, as an another step in their ongoing study (Refs. 50-61). The literature review and a description of the problem are presented in great detail and deeply acquaint the reader with the topic. I have only few minor comments/questions:
What is the point of using M06 XC for single-point calculations? The numbers appear in Table 1, but there is no any explanation or comparison to B3LYP-D2. At least short comment would be great.
Authors’ reply – We are grateful to the reviewer for the careful reading and very positive comments. Comments of the referee have been addressed adding new paragraphs in the manuscript.
Author action – We add the following sentences:
In Section 2, at lines 155-162, “In fact, the reliability of the semi-empirical DFT-D2 approach as applied to the systems of interest here can be questionable to some extent. The highly parameterized forms of hybrid meta-GGA exchange–correlation functionals, the M06 suite is designed, for application in the area of non-covalent interactions and transition metal bonding. These DF hamiltonians are specifically suited to predict energy estimates for highly correlated systems featuring transition metals and non-bonded interactions. Therefore for a better evaluation of the energetics discussed in this work M06 estimates have been also considered.”
In Subsection 3.2, at lines 258-262, “Table 1 shows that the differences in M06 and B3LYP-D2 adsorption energies are negligible with the only exceptions for D, G, I minima i.e. the adsorption processes involving the TiCl4EB complex where M06 estimates are significantly larger. That may be attributed to the lower cost predicted, at M06 level, for the deformation of the complex on cluster respect to the molecule in gas phase.”
In Subsection 3.4 at lines 425-430, “Similarly to what happened in the case of adsorption processes, we found that a surface reaction starting from MgCl2/TiCl4...EB (Model B) and leading to MgCl2/TiCl4 (EB) (model E) is energetically not favored by ~61 kJ/mol in M06 case against 35 kJ/mol predicted by Grimme method; whereas the evolution to the homogeneous like TiCl4(EB)2 surface adduct is disfavored by 68 kJ/mol at M06 level against 55 kJ/mol in B3LYP-D2 case.”
2) Comment – You show that absorption leads to various changes in vibrational spectra, and also assign those modes to specific atoms vibrations. But the explanation for this changes is very brief. I wonder, why don't you analyze charge redistribution/transfer for adsorbates as it is crucial for catalysis?
Authors’ reply – As we thoroughly described in a previous paper of us (see Ref.[57] of the present manuscript), the changes in vibrational spectra of EB i.e. for typical CO stretching can be attributed to the polarization effects on CO originating from more or less acidic sites of the nanoclusters and/or the perturbations coming from nearby TiCl4; in a reciprocal way, changes occur in low frequency range for the TiCl4 counterpart for the same reason.
The effect of the presence of donors on charge density and charge transfer/redistribution on catalysis is a topic that we are investigating with different methodologies but is out of the aims of the present paper. That being said, Mulliken analysis has been performed on models we considered in the manuscript; for selected minima (B, L), where the effect of co-adsorption are expected to be more relevant, we introduced some comments we report hereafter.
Author action – We introduced in the manuscript:
In Subsection 3.2, at lines 263-279, “Mulliken analysis has been performed on models. In particular, we considered how Mulliken charges of Ti and of the two Cl atoms bound exclusively to Ti (free chlorine atoms exposed to reaction medium) of adsorbed MgCl2/TiCl4 are modified in the presence of one or two co-adorbed EBs as in models B and L, respectively. In model B we find the following charges: Ti (+1.567 |e|), Cl (-0.362 and -0.286 |e|), whereas in model L we got: Ti (+1.579 |e|), Cl (-0.386 and -0.286 |e|); corresponding values for MgCl2/TiCl4 are for Ti (+1.547 |e|) and for Cl (-0.268 and 0.286 |e|). Even between the limits of a Mulliken treatment, we can infer that the presence of EB (1 or 2 molecules) increases the charge on Ti and also the difference between the charges on the two Cl atoms bound to it, thus enhancing the not equivalency of the two Ti-Cl bonds that could be relevant in the catalytic process starting with the cleavage of one Ti-Cl bond. Nevertheless, this suggestion should be more properly investigated by considering the electronic effect of donors in presence of the reduced form of Ti i.e. (Ti(III)) that is supposed to be the active species in Ziegler-Natta polymerization catalysis ([82],[83]. The effect of the presence of donors on charge density, charge transfer/redistribution and on catalysis is a topic that we are investigating with different methodologies but is out of the aims of the present paper.”
3) Comment – E, H, F and L models in Table 1 have lowest energies, but they are incomparable to other models due to different number of adsorbates, as well you describe in the text where the excess of energy comes from for EB for example. Maybe it is possible to put energies per adsorbed EB into Table 1 or find another way to present the results of thermodynamics?
Author reply – Table 1 reports what we can define the energy of a reaction, in the case of the models indicated by the referee we report the energy variations associated to the adsorption processes. We do not believe that a different way can be adopted to report the energy of adsorption.
4) Comment – Multiple times in the text you use surface indices without brackets, it can be confusing to the reader.
Author reply – We thanks the reviewer for careful reading and we modified the text according to his minor concerns.
Author action – We introduced the modifications:
We added round brackets to indicate the family of planes all along the manuscript.
Reviewer 3 Report
In this paper, the authors describe the effect of adsorption and binding of the electron donor ethyl benzoate on Ziegler-Natta polymerization catalytic nanosystem by DFT calculations. The investigation was performed with nanopalates of 50 MgCl2 and 50MgCl2/3TiCl4. The authors have used simulations of vibrational and Raman spectroscopy to identify the binding process or the formation of homogeneous-like catalytic species. The results reported in this paper are important and can give new opportunities for designing proper polymerizations catalysts. The paper can be published after the following corrections:
- Separate the section of Discussion and Conclusions. It is much better to have a separate section for conclusions that summarizes also the main findings.
- Correct the following typos:
Line 14: replace “heterogenous” by “heterogeneous”
Line 14: replace “determing” by “determining”
Line 50: replace “aspecific” by “specific”
Line 62: replace “analize” by “analysis”
Line 126: replace “wheter” by “whether”
Line 225: replace “antysimmetric” by “antisymmetric”
Line 321: replace “tinily” by “tiny”
Line 375” replace “homogeneus” by “homogeneous” and “unumbiguosly” by “unambiguously”
Author Response
We thank the Reviewer 3 for his comments and useful suggestions received by our manuscript.
In the revised version, we have responded to the concerns raised by the reviewers as outlined in the present letter. In the following, we report detailed point-by-point reply to your comments for your convenience, the following colour code has been used: blue and italic: comments of the reviewer; black: our replies to the reviewer’s comments; red: modifications that were applied in the text to address the indications of the reviewer.
We think that this new amended version can be suitable for publication in Materials.
Torino, 17th January 2022.
Yours sincerely,
Maddalena D’Amore (on behalf of all the authors).
Reviewer: 3
1) Comment – In this paper, the authors describe the effect of adsorption and binding of the electron donor ethyl - benzoate on Ziegler-Natta polymerization catalytic nanosystem by DFT calculations. The investigation was performed with nanopalates of 50 MgCl2 and 50MgCl2 /3TiCl4 The authors have used simulations of vibrational and Raman spectroscopy to identify the binding process or the formation of homogeneous-like catalytic species. The results reported in this paper are important and can give new opportunities for designing proper polymerizations catalysts. The paper can be published after the following corrections: Separate the section of Discussion and Conclusions. It is much better to have a separate section for conclusions that summarizes also the main findings. Correct the following typos:
Line 14: replace “heterogenous” by “heterogeneous”
Line 14: replace “determing” by “determining”
Line 50: replace “aspecific” by “specific”
Line 62: replace “analize” by “analysis”
Line 126: replace “wheter” by “whether”
Line 225: replace “antysimmetric” by “antisymmetric”
Line 321: replace “tinily” by “tiny”
Line 375” replace “homogeneus” by “homogeneous” and “unumbiguosly” by “unambiguously”
Authors’ reply – We thanks the referee for the accurate revision and very positive evaluation of manuscript. We separated the discussion section from the conclusions, namely summary and perspectives. All the listed typos have been replaced according to reviewer’s suggestions, except the “aspecific” that is right.
Round 2
Reviewer 1 Report
The manuscript was significantly improved, and the authors successfully addressed the questions and comments from the previous review. Therefore, this new version is suitable for publication in Materials.